# Unlocking the Potential of Sprouted Cereals, Pseudocereals, and Pulses in Combating Malnutrition

**DOI:** 10.3390/foods12213901

**Published:** 2023-10-24

**Authors:** Mahsa Majzoobi, Ziyu Wang, Shahla Teimouri, Nelum Pematilleke, Charles Stephen Brennan, Asgar Farahnaky

**Affiliations:** Biosciences and Food Technology, RMIT University, Bundoora West Campus, Plenty Road, Melbourne, VIC 3083, Australia; s3742701@student.rmit.edu.au (Z.W.); shahlateimouri2@rmit.com (S.T.); nelum.pematilleke@rmit.edu.au (N.P.); charles.brennan@rmit.edu.au (C.S.B.); asgar.farahnaky@rmit.edu.au (A.F.)

**Keywords:** germination, nutrient-dense, environmentally friendly, grains, functional foods

## Abstract

Due to the global rise in food insecurity, micronutrient deficiency, and diet-related health issues, the United Nations (UN) has called for action to eradicate hunger and malnutrition. Grains are the staple food worldwide; hence, improving their nutritional quality can certainly be an appropriate approach to mitigate malnutrition. This review article aims to collect recent information on developing nutrient-dense grains using a sustainable and natural process known as “sprouting or germination” and to discuss novel applications of sprouted grains to tackle malnutrition (specifically undernutrition). This article discusses applicable interventions and strategies to encourage biochemical changes in sprouting grains further to boost their nutritional value and health benefits. It also explains opportunities to use spouted grains at home and in industrial food applications, especially focusing on domestic grains in regions with prevalent malnutrition. The common challenges for producing sprouted grains, their future trends, and research opportunities have been covered. This review article will benefit scientists and researchers in food, nutrition, and agriculture, as well as agrifood businesses and policymakers who aim to develop nutrient-enriched foods to enhance public health.

## 1. Introduction

Grains, including cereals, pseudocereals, and pulses, are staple foods worldwide and significantly contribute to human nutrition and well being [1]. The popularity of grains has further increased with the growing demand for sustainable plant-based foods. However, over the last century, selected grains with high production yields have been grown to provide affordable foods for the world’s growing population and to combat food security risks due to climate shocks and disruption of the food supply chain. This has raised major concerns since selective breeding has drastically reduced the diversity of the available grains for food consumption [2]. The dominance of highly productive grains known as leading crops (e.g., wheat, rice, and corn) has marginalized the production and utilization of other grains known as minor grains (e.g., pseudocereals). In addition, by increasing the production yield of leading grains, their nutrient concentration, especially protein and micronutrients, has diminished significantly [3]. Other factors, including unsuitable soil conditions and climate shocks, can cause poor nutrient accumulation in the grains [4]. As a result, greater consumption of grains is required to provide the Food and Drug Administration (FDA) recommended intake of nutrients leading to the increased risk of diet-related noncommunicable diseases, including obesity, type 2 diabetes, cardiovascular diseases, and micronutrient deficiency [2,5]. The global rise in micronutrient deficiency and diet-related health issues coincide with the increased consumption of high-yielding crops over the last few decades [2,6].

According to the World Health Organization (WHO), “malnutrition” refers to deficiencies or excesses in nutrient intake, imbalance of essential nutrients, or impaired nutrient utilization and consists of undernutrition, overweight, obesity, and diet-related noncommunicable diseases. One of the common forms of malnutrition is the lack of protein, minerals, and vitamins in a diet, which are essential for healthy growth and body functions that affect both developed and developing countries [7]. Globally, more than 2 billion people (one in three) experience micronutrient deficiencies, also known as a “hidden hunger”, which is highly prevalent among women, children, and the elderly [4]. According to the Global Report on Food Crises (GRFC) 2023, more than 250 million people in 58 food-crisis countries/territories required immediate food assistance in 2022. This represents the highest recorded figure in the seven-year history of the Global Report of Food Crises (GRFC) [8].

Due to the increase in hunger, malnutrition, and diet-related noncommunicable diseases, the United Nations (UN) has declared the decade (2016–2025) one of action on “Nutrition” to eradicate hunger and malnutrition and to diminish the burden of diet-related health issues in all forms and for all age groups. To tackle malnutrition, the UN has defined six areas of action as led by the WHO and the Food and Drug Administration (FDA) (Figure 1). One of the policy actions of the UN to tackle malnutrition is “creating sustainable, resilient food systems for healthy diets” [7].

In line with this area of action, this review article aims to compile recent insights into developing nutrient-rich grains through an eco-friendly and natural process known as “sprouting” or “germination”. Additionally, it explores the potential of sprouted grains as a viable solution to address malnutrition, particularly undernutrition. Furthermore, it delves into the possibilities, obstacles, and prospects associated with their wider use in developing foods rich in protein, calories, vitamins, and minerals.

## 2. Sprouting Process

Sprouting, also known as germination, is a historical process practiced since 5000 years ago. The sprouting process is a natural, flexible, cost-effective, and sustainable process of grain enrichment with macro and micro-nutrients required to treat malnutrition. When performed under optimum conditions, it is an effective, sustainable, and green intervention to enhance the nutritional quality and health benefits of the grains and to reduce/eliminate some of their major limitations, such as anti-nutrients, allergenicity, and indigestibility. It is also highly effective in mitigating some grain processing complications such as dehulling, long cooking time, and undesirable taste [9,10,11]. Sprouted grains are whole grains that have just begun germinating and are collected before sprout growth exceeds kernel length [12].

The sprouting process is adoptable at home, traditional, and industrial settings and can be applied to various grains, including native grains of areas with a prevalence of undernourishment (e.g., sorghum and pseudocereals), as well as modern grains (e.g., rice and wheat) to transform them into nutrient-dense products in a short time [13,14].

The sprouting process initiates when the grains are exposed to water and accelerates with increasing temperature. The process involves three major phases, shown in Figure 2. The first phase, known as “imbibition”, occurs during the soaking or steeping of the grains in water (often 1:1.5 *w*/*v*). It involves rapid water uptake and full hydration of the dry seeds. The absorbed water travels across the kernel and reaches the germ, scutellum, seed coat, and pericarp. This stage may take about 24 to 48 h at about 20–25 °C, during which the steeping water is often replaced to avoid contamination. In the second phase, known as “sprouting” or “germination”, the grains are drained and transferred into sprouting vessels and incubated under controlled humidity and temperature depending on the grain type, genetic variation, source, and age. However, a relative humidity of about 85–99% and a temperature range of 20–30 °C for about 3–7 days with or without light are commonly used. At this stage, the grains rapidly restore their metabolic activity and release or synthesize abscisic acid, gibberellic acid, and ethylene as the main plant hormones that stimulate the production of enzymes, including cellulase, phytase, amylase, protease, and lipase to produce simple molecules (e.g., glucose, amino acids, and phosphorous) required for the embryo to grow. The activity of these enzymes brings about significant changes in the biochemical composition, nutritional quality, health benefits, and organoleptic properties of the grains [12,15].

## 3. Biochemical Changes

During sprouting, some biochemical changes occur that enhance the nutritional quality of the grains to address undernourishment and its consequent health issues. Previous studies on the sprouted grains have indicated that the extent of biochemical changes in the sprouted grains is controlled by various parameters, including grain species and variety, sprouting conditions, the seedling growth stage, and laboratory techniques [16,17]. Additionally, different biocomponents of the same grain may vary differently or remain unchanged under similar sprouting conditions; thus, choosing a sprouting condition that favors some components may be against others. Thus, the exact nutritional composition of sprouted grains is unclear and may exhibit various biological health effects compared to ordinary grains or different sprouting conditions [16,18,19]. Table 1 shows sprouting conditions that positively affect increasing nutrients and bioactive compounds in the grains.

### 3.1. Starch and Sugars

Starch is the main carbohydrate in grains (65–80%) and provides more than half of all calories in the human diet. It is essential to provide energy for daily activities and body functions, and the lack of energy in the diet leads to energy malnutrition [20]. During the sprouting process, starch is rapidly hydrolyzed by the action of α- and β-amylases and α-glucosidase into dextrins and simple sugars, resulting in improved starch digestibility and higher values of the glycaemic index (GI) [12,15]. The indigestible part of starch (resistant starch) remains mainly intact during germination [21,22]. Some studies have shown increased starch content during a prolonged soaking process due to the leaching of water-soluble components such as proteins and sugars, as reported for wheat, barley, and some lupin varieties soaked at 35 °C for 24 h [23].

The sugars produced during sprouting are used as a source of energy for the developing embryo. The dominant type of accumulated simple sugars in the grains is defined by grain species and the level of starch-hydrolyzing enzymes. For instance, in chickpeas and green peas, the main accumulated sugar is sucrose; in buckwheat, it is glucose, while in millet, maltose is the dominant sugar [24,25].

### 3.2. Fiber

Whole grains are rich sources of dietary fiber (up to 18%), which is essential for the healthy function of the digestion system, increasing satiety, and reducing the risk of obesity, diabetes type 2, cardiovascular diseases, and some cancers [26]. Short sprouting time (~72 h) can enhance the total fiber and insoluble content of brown rice primarily due to the formation of new cells, the loss of other compounds such as carbohydrates and proteins due to respiration and leaching, and the dissociation of fiber from other molecules due to the hydrolysis process [27]. However, the reduction in the soluble dietary fiber by increasing the sprouting time (about 96 h) has been related to the excessive breakdown of this molecule, especially beta-glucan and arabinoxylans. By increasing the sprouting time (~96 h), the enzymatic hydrolysis increases, resulting in the excessive breakdown of the fiber and lower fiber content. Other reports indicated no change in the total fiber content of sprouted barley and rice [27,28]. The content of insoluble fiber, however, increased by applying long sprouting periods (2–6 days), as shown for wheat, oat, and barley malts [28,29].

### 3.3. Proteins and Amino Acids

As per the WHO recommendations, the recommended dietary allowance (RDA) for protein is 0.83 g per kilogram of body weight per day [20]. However, protein deficiency is highly common around the world, resulting in one of the most prevalent forms of malnutrition manifested as growth failure, loss of muscle mass, decreased immunity, weakening of the heart and respiratory system, and even death [20,30]. Grains can, therefore, be considered nutritious, healthy foods that raise the nutritional effectiveness of the malnourished majority in the developing parts of the world. Grains, especially pulses, are considered rich and sustainable sources of plant proteins (~7–30%) [1]. The protein content of the sprouted grains depends on the balance between protein degradation and protein biosynthesis during sprouting. For some grains, such as sorghum, lentil, and horse gram, protein content decreased due to the protein hydrolysis to peptides and amino acids by protease [31]. However, some varieties of lentils showed no change in protein content after sprouting. At the same time, some reports indicated a slight increase in the protein content by increasing the sprouting time, which has been related to the reduction or loss in carbohydrates due to enzymatic degradation or leaching during soaking and also the release of bound proteins and amino acids [10]. In wheat, barley, triticale, rye, and oats, the prolamins content decreases by increasing the sprouting time. This can positively reduce grain allergenicity since prolamins are the main allergens in these grains [29].

Amino acids such as tryptophan, leucine, lysine, and valine are essential in the production of proteins in the body that help in nutrient absorption, tissue growth, immune function, and energy production. Grains, especially oats and quinoa, are rich in most amino acids but relatively low in lysine, threonine, leucine, and histidine [32].

Bioactive peptides are known as protein pieces made up of 2–20 amino acids and exhibit positive biological benefits to prevent or treat some chronic diseases such as anti-inflammatory, antimicrobial, anti-obesity, antioxidant, and antihypertensive attributes [33]. In most grains, an increase in amino acid and bioactive compound content has been observed due to proteolytic activity during sprouting, which is desirable for the improvement of the nutritional quality of the grains. The type of liberated amino acid varies depending on the grain type of sprouting conditions. In sprouted brown rice, the content of most amino acids increased, except histidine, methionine and threonine, glutamic acid, aspartic acid, and serine [32]. In sprouted barley, the content of isoleucine, leucine, lysine, methionine, phenylalanine, and tryptophan increased significantly. In cereals, protein hydrolysis leads to the hydrolysis of prolamins, and the liberated amino acids, such as glutamic and proline, are converted to the limiting amino acids, such as lysine, leading to improved protein quality [32].

#### γ-Aminobutyric Acid (GABA)

The GABA is a four-carbon non-protein amino acid generated by the decarboxylation of L-glutamic acid catalyzed by the glutamate decarboxylase enzyme. It is the primary inhibitory neurotransmitter in the mammalian brain and can be found naturally in many foods, including grains [17]. Its numerous health benefits include regulating blood pressure, controlling nerve cell hyperactivity, reducing stress and anxiety, and relieving pain [34,35].

A significant increase in GABA content has been reported for many sprouted grains, including soybean, black bean, white and colored rice, and chickpeas [36,37]. GABA accumulation was reported to be initiated during the soaking process and varies with the varieties, pH of the soaking water, soaking time, and temperature. The increase in GABA content in white and colored rough rice has been reported to have a positive correlation with sprouting time and has been correlated with the decomposition of storage protein into amino acids and the conversion of glutamic acid to GABA through the glutamate decarboxylase enzyme [37,38].

### 3.4. Lipids and Fatty Acids

Lipids and fatty acids are important components of a balanced diet. They are critical for cell structure, function, energy, organ and body insulation, and protection. Grains generally contain 2–6% of lipids mostly found in the germ, scutellum, and aleurone layer mostly in the form of triglycerides or triacylglycerols [34,39]. During sprouting, lipids are hydrolyzed by the action of lipases into free fatty acids, which increase at higher temperatures. Free fatty acids are then converted to sugars through β-oxidation and the glyoxylate cycle and generate energy and carbon for biochemical and physicochemical changes of the sprouting grains [40]. In a prolonged sprouting process, the increase in lipase and lipoxygenase levels results in off-flavor due to the formation of free phenolic compounds, aldehydes, dimethyl sulfide, and heterocyclic. In chickpeas, green peas, and lentils, up to 35%, 19%, and 10% lipids were lost, respectively, compared to their unsprouted counterparts [40]. Various results have been reported for the lipid content of sprouted grains. The lipid content increased in sprouted soybeans, sorghum, and oats, while it decreased in sprouted millets but remained unchanged in sprouted chickpeas, green peas, mung bean, faba bean, black bean, wheat, and barley [10,23,41].

Fatty acids play a pivotal role in human health. It is essential for various physiological functions of the body. Inadequate intake of fatty acids can lead to growth retardation, impaired immune function, neurologic disorders, and skin abnormalities [31].

Sprouting alters the fatty acid content and composition of the grains. In sprouted wheat grains, cis-18:1, cis, and cis-18:2 decreased, but 18:3 n3 (omega 3) content increased. However, it had no significant effect on the fatty acid composition of both free and bound lipids of waxy wheat, where the major components are polyunsaturated fatty acids, followed by saturated and monounsaturated fatty acids [39].

### 3.5. Minerals

Minerals are the essential micronutrients required for human health, which play a crucial role in various metabolic processes and the biosynthesis of macronutrients. Grains are rich in some minerals such as iron (Fe), zinc (Zn), copper (Cu), selenium (Se), and magnesium (Mg); however, the bioavailability of these minerals is low due to the chelating effects of phytic acid and hence consumption of the common grains cannot provide the required amount of minerals [21,42].

Mineral deficiency is a common form of malnutrition, predominant in low-income families, young women, children, and the elderly. The most associated minerals with micronutrient malnutrition are Zn, Se, Fe, and Ca [43]. Sprouting has been considered a successful method to increase the content and bioavailability of Ca, Fe, Zn, Mg, and Se, as reported for sprouted corn, chickpeas, mung beans, cowpeas, and black beans [44,45]. The evident rise in mineral content may have been caused by the loss of fat and carbohydrate contents during sprouting. The increase in mineral bioavailability is due to the increased phytase activity during the sprouting process, which in turn degrades phytates and releases the minerals [21,34]. It appears that sprouting time has a significant impact on the mineral content and its bioavailability in the grains. For instance, in corn, the mineral content initially reduced after 2 days of sprouting and then increased, followed by 6 days of sprouting [12]. For sprouted sorghum, no significant difference in mineral content has been observed; a decrease in mineral content in some sprouted grains such as chickpea, lentil, barley, soybean, mung bean, faba, and black bean has been reported during a prolonged sprouting process due to leaching or their use in other biochemical reactions required for kernel development [10,12,21].

### 3.6. Vitamins and Bioactive Compounds

Vitamins are a chemically diverse group of organic compounds classified into water-soluble vitamins (vitamin C and B-group) and fat-soluble vitamins (A, D, E, and K). Vitamins are essential for normal body functions and growth. However, the human body cannot synthesize or store most of the vitamins (especially water-soluble vitamins) in an adequate amount; hence, vitamin intake through diet is essential. Despite the importance of vitamins in the diet, vitamin deficiency is highly common around the world [46].

Whole grains have a significant contribution in providing vitamins B and E [47]. The vitamin content of the grains can be further increased by sprouting [48]. Sprouting emerges protective responses through the synthesis and release of bioactive compounds, antioxidants, vitamins, and phenolic compounds that are highly necessary for healthy body functions [49,50,51]. Grains are rich sources of B vitamins (riboflavin, thiamine, pyridoxine, and niacin) and tocopherols. Several studies have shown an increase in the vitamin content of sprouted grains such as wheat, oats, barley, and rice mostly due to the reactivation of vitamin biosynthesis during sprouting [17,52]. The influence of sprouting on vitamin content depends on the grain genotype and sprouting condition. For instance, the tocopherol content increased in soybeans but reduced in mung bean and lupin [53]. In chickpeas, a decline in tocopherol and beta-carotene content was observed, which was related to their role in preserving lipids from oxidation during sprouting [51]. Although grains are not considered a source of vitamin C, some reports indicated an increase in this vitamin upon the sprouting of barley and wheat, especially when exposed to light during germination [12].

Bioactive compounds are present in small quantities in foods, mainly in fruits, vegetables, and whole grains, and provide numerous health benefits. High consumption of foods rich in bioactive compounds with antioxidant activity, including vitamins, phytochemicals, and mainly phenolic compounds, such as flavonoids and carotenoids, has a positive effect on human health and could diminish the risk of numerous diseases, such as cancer, heart disease, age-related functional decadence, and malnutrition [13,54].

Antioxidants produced during sprouting as the primary and secondary metabolite compounds have numerous documented health benefits, including anticancer effects, blood cholesterol and blood pressure lowering effects, antidepression, memory and emotional regulation, prevention of diabetes type 2 and its complications, and reducing the risk of obesity [54].

Bioactive compounds (phenolic compounds, flavonoids, carotenoids, and melatonin) and the antioxidant activity of the grains often increase during sprouting. The increase in the polyphenol content can be due to the activation of the highly active enzyme phenylalanine ammonia–lyase that encourages the biosynthesis of polyphenols. In addition, enzymatic hydrolysis of polyphenols–carbohydrates or polyphenols–proteins has been shown to release the bound phenolic compounds from the cell walls and better extractability [11,34,55]. Loss of other components during sprouting, such as carbohydrates and proteins, can also increase total phenolic compounds [56]. It has been reported that the content of the bioactive compounds varies in sprouted soft and hard wheat cultivars. The hard wheat showed the highest increase in antioxidant capacity after sprouting [57]. The antioxidant activity of amaranth, quinoa, and buckwheat increased significantly upon sprouting. Total phenolic content was doubled following the sprouting of quinoa, buckwheat, and wheat and quadrupled in the case of amaranth. Under similar sprouting conditions, buckwheat had the highest total phenol content, followed by quinoa and amaranth [58].

Although most studies show increased bioactive compounds in the sprouted grains, some contradictory results have been reported. The sprouting of chickpeas has been reported to reduce the carotenoid content by approximately 60% after 48 h of sprouting compared to its unsprouted counterparts, which has been related to the degradation of carotenoids to prevent lipid oxidation [10,51]. A similar decrease trend in carotenoid content was reported in sprouted cowpeas, Bambara nuts, pigeon peas, and groundnuts, and the percentage of decrease is associated with an increase in sprouting time [59]. In addition, a decrease in antioxidant activity has been reported in sprouted black beans, mung beans, and lentils, and it could be due to the genetics of the legume, which influences the phenolic compound synthesis during sprouting [34]. In sorghum, soaking for 24 h at room temperature in the dark, followed by sprouting for 3 days, had a negative impact on in vitro antioxidant activity and an antihypertensive effect but improved erythrocyte protection. The non-sprouted sorghum had better nutraceutical potential; however, the sprouting could positively impact the profile of bioactive compounds involved in the protection of human erythrocytes from oxidative damage. The reasons for such changes are not fully known in different grains [60].

### 3.7. Anti-Nutritional Factors

Grains contain natural anti-nutrients, including phytates, trypsin inhibitors, tannin, lectin, oxalate, and saponin, which can not only reduce the bioavailability and digestibility of the nutrients but also cause a bitter taste. Sprouting has been found to reduce different types of anti-nutrients in grains, as reported for many grains, including sorghum, quinoa, rice, barley, and wheat [61,62]. However, most of the available studies have focused on phytic acid as the most effective anti-nutrient in grains since it makes complexes with most minerals such as iron, calcium, zinc, and magnesium and reduces their absorbance rate and bioavailability in the body, resulting in a mineral deficiency [63].

Sprouting is a successful treatment and the most-used strategy to reduce phytic acid in grains. The activity of the phytase enzyme during sprouting degrades phytic acid to release the phosphorous required for sprouting. The decrease in the phytic acid content results in an increase in mineral bioavailability. A phytate reduction of 51% in sprouted wheat that was hydrothermally processed increased [63,64,65]. The effects of sprouting on anti-nutrients depend on the species and sprouting conditions. A phytic acid reduction of 98% in oats, 84% in rye, 58% in barley, 4–60% in brown rice, and 63% in wheat has been reported after sprouting. The reduction of phytic acid in sprouted grains increases the bioavailability of several minerals, which leads to the increased nutritional value of the food products. Some studies have shown an increase in total tannin and saponin content reported in sprouted chickpeas, buckwheat, and soybeans [21,66,67].

### 3.8. Toxic Compounds

Grains are highly susceptible to contamination by molds and their toxins, known as mycotoxin, which are natural contaminants and toxic metabolites produced by various fungal species. Sprouting has been reported as an effective method to reduce about 40–63% of mycotoxins in wheat sprouted at 25 °C and ~85% relative humidity for about 10 days [63]. A significant reduction in aflatoxins B1, B2, G1, and G2 during sprouting has been observed and related to an increase in trace elements (Ca, Mg, Fe, and Zn content), which can affect the enzymatic system involved in aflatoxin degradation. Sprouting activates aflatoxin-degrading enzymes such as biphenyl-dioxygenases, dihydro-diol-dehydrogenases, and hydrolases that convert aflatoxins into substances with no or less toxicity [63]. However, the effects of prolonged sprouting (about 10 days) detoxification on the nutritional quality of the grains were not investigated in this research.

**Table 1 foods-12-03901-t001:** Sprouting conditions and biochemical changes that are in favor of improving the nutritional quality of some grains.

Sprouted Grain	Condition Used	Finding	Ref.
Normal and waxy wheat	Soaking: 20 °C, 6–8 h Sprouting: 20 ± 5 °C, 9–11 days on soft agar (1.5% agar)	Increase in protein content and digestibility, fiber, ash, total phenolic and flavonoid contents, and antioxidant activity, anti-inflammatory effects. Decrease in fat, carbohydrate, phytic acid, and cell toxicity.	[68,69,70]
Brown rice	Soaking: 28–30 °C, 12 h Sprouting: 24 h, 28–30 °C, 90–95% RH	Increase in protein content and digestibility, starch digestibility, niacin, amino acids, vitamin E, vitamin C and pyridoxine, GABA, γ-oryzanol, antioxidants. Decrease in phytic acid. No change in ash, crude fat, or carbohydrate.	[70,71]
Mung bean	Soaking: 25 °C, 12 h Sprouting: 60 h, 25 °C, 70% RH	Increase in protein content and digestibility, ash, crude fiber. Reduction in carbohydrate, resistant starch, and crude fat.	[72]
Chickpea	Soaking: 25 °C, 12 h Sprouting: 60 h, 25 °C; 70% RH	Increase in protein, ash, crude fiber, and reduction in carbohydrate and crude fat.	[73]
Quinoa	Soaking: 22 °C, 4 h Sprouting: 72 h, 22 °C, 95% RH	Increase in protein, ash, and crude fiber and reduction in carbohydrate, crude fat, and ash.	[74]
Buckwheat	Soaking: 22 °C, 1 h Sprouting: 12, 24, 36, 48, 60, and 72 h, 25 °C, 90% RH	Increase in protein, total phenolic, and flavonoid content. Decrease in crude fat and phytic acid.	[75]
Oat	Soaking: 22 °C, 4 h Sprouting: 18 °C for 96 h, ≥90% RH, in darkness	Increase in reducing sugar, protein, essential amino acids, minerals (Ca, Fe, Zn, Mg), riboflavin, and essential fatty acids. No change in soluble and insoluble fiber.	[76]

## 4. Processing of the Sprouted Grains

Figure 3 shows an overview of the food applications of the sprouted grains. Fresh sprouts are commonly used as culinary ingredients in restaurants and homemade cooking. Although rich in nutrients and bioactive compounds, they are prone to microbial contamination and rapid spoilage if not properly handled, packaged, and stored. Additionally, fresh sprouts are rich in some enzymes (e.g., amylase, protease, and lipase) and biological compounds (e.g., reducing sugars, amino acids, unsaturated fatty acids, and phenolic compounds). During food processing and storage, these compounds can contribute to some undesirable chemical reactions such as Maillard reactions, oxidation, and starch and protein hydrolysis, resulting in adverse effects on the quality and shelf-life of the end products. Thus, fresh sprouts are often further dried using oven, solar, or freeze-drying methods to increase their shelf-life and deactivate enzyme activity [77].

Dried and powdered sprouted grains have been included in healthy snacks, bread, breakfast cereals, noodles and pasta, prebiotic foods, fermented products, and baby foods [61,62]. However, it is critical to assess the nutritional quality of the end products to convey the expected health benefits to the consumers. Table 2 shows some food products formulated with different sprouted grains and the observed changes in quality, nutritional value, and health benefits. Bioactive compounds and many biochemical compounds such as proteins and lipids are often lost during food processing and storage as they are often decomposed at elevated temperatures, exposure to light, high shear, and pressure. Thus, choosing the suitable processing type and conditions to protect nutrients in the foods are of great importance.

The use of sprouted wheat and other grains such as pseudocereals, rice, and lentils in bread-making has been reported, which could manipulate obesity and diabetes in mice [78]. Nevertheless, these studies have indicated that using sprouted grains in bread-making requires careful control of the sprouting conditions and bread-making process. In a prolonged sprouting process, excessive production of hydrolyzing enzymes such as alpha-amylase, similar to the preharvest sprouting, can result in inferior dough and bread quality. It has been indicated that a sprouting time of 12 h is enough to enhance the nutritional quality of wheat without triggering excessive hydrolyzing enzymes [61]. Most of the previous studies indicated an accelerated fermentation process of the dough, a darker crust color, a softer bread crumb, a sweeter taste, and a longer shelf-life of the bread. Similar results have been reported when other sprouted grains, including oats, brown rice, maize, barley, and their mixture have been used in bread production [62,79].

Sprouting has been used to reduce grain hardness as a common issue in processing, for example, in brown rice cooking. It creates some micro-cracks and features on the grains, which can facilitate water penetration into the grains, resulting in shorter cooking time, which is an issue with hard grains such as pulses. It has been used to reduce the long cooking time of some grains, such as brown rice and barley; however, the stickiness of the cooked rice remains the same [15].

Sprouted grains are also suitable for the production of special-diet foods. Using sprouted pseudocereals such as buckwheat as a natural way of enhancing the nutritional quality of gluten-free bread has been reported. The resulting bread had significantly higher antioxidants and phenolic content [15].

It has been reported that sprouting can be used to reduce oligo-, di-, monosaccharides, and polyols (FODMAPs) compounds in some grains, including pulses and pseudocereals, to make them suitable for a low FODMAP diet. However, this method is not suitable for cereals as it accumulates fructan in the grains [80].

**Table 2 foods-12-03901-t002:** Addition of some sprouted grains in foods and their effects on quality, nutritional value, and health benefits.

Spouted Grains	Sprouting Conditions	% Sprouted Grains in Foods	Product Quality	Nutritional Value of the Product	Health Benefits	Reference
Lentil	Soaking: 24 h at 20 °C Sprouting: 96 h, 90% RH, 25 °C	0, 10, 20% in bread	Up to 10% volume increased but reduced at 20% replacement; Darker color; Harder texture.	10% reduced phytic acid; 20% increase in iron; 10% increase in protein; no change in antioxidant content	Not reported	[81]
Wheat	Soaking: 26 °C, 6 h. Sprouting: 20 ◦C, 80% RH, 12 and 24 h	10 to 50% in bread and wheat-based fermented beverage	Slower starch retrogradation. Sensory attributes increased	Not reported	Not reported	[82]
Wheat, barley, pearl millet, and green gram	Soaking: in water, 8 h, 30 °C. Spouting: 35 °C, 95% RH, 24–36 h	2–8% in non-dairy probiotic drink	Increased Sensory acceptability and probiotic count	Increase in protein digestibility	Not reported	[83]
Corn	Soaking: 8 h Sprouting: 72 h, 24 °C 80–90% RH	Non-alcoholic fermented beverages (kombucha)	Improved appearance, flavor, smell, color, and mouthfeel	Not reported	Improved health benefits	[84]
Buckwheat, ed Job’s tears, and mungbean	Soaking: 10–24 h, room temperature Sprouting: 6 h-7 days, 10 °C	20% in rice cakes	Decreased starch pasting properties. Slowing retrogradation; Improved textural and sensory properties	The dietary fiber increased. Texture properties and sensory evaluation improved.	Improved starch digestion	[14]
Brown rice	Soaking: in water, 24 h at 28 °C Sprouting: 48 h and 96 h	Fermented yoghurt	Promoting the growth of the starter culture in yogurt. Increase in the average scores for organoleptic properties	Increase in phenolic compounds and GABA, consistency index, and density	Low glycaemic index, increase in health-promoting properties	[85]
wheat barley, chickpea, lentil, and quinoa grains	Soaking: in water, room temperature, 30 min Sprouting: in water, 24 h, 16.5 °C	20% in bread	Improved sensory quality	Increase in peptides, free amino acids, GABA, Decrease in phytic acid, tannins, raffinose, and trypsin inhibitors.	High protein digestibility and low starch availability	[86,87,88]

## 5. Challenges of Using Sprouted Grains in Foods

Fresh sprouts are highly prone to biological, chemical, and physical contaminations that occur during pre- and post-harvest processing [89]. Fresh sprouted grains are increasingly perceived as potentially hazardous foods contaminated mostly by the main sources of food-borne pathogens such as *Escherichia coli* O157, many serotypes of *Salmonella* and *Bacillus cereus*, and pathogens and dangerous filamentous fungi such as *Aspergillus clavatus*, *A. flavus*, *A. niger*, *Fusarium nivale*, *F. culmorum*, *Trichothecium roseum*, and *Penicillium* spp. [89,90]. Thus, all stages of sprouting, from seed selection to the consumption stage, must be under control to prevent any food safety hazards. Strategies to eliminate microbial risks have been described previously [91].

Another challenge is related to the high dependency of grain nutrients on various internal and external factors such as grain type and variety, physiology and dormancy, and variation in the sprouting and post-processing conditions, leading to inconsistency in the reported biochemical changes and health benefits [42,92]. Additionally, the conditions with positive effects on some biochemical components may not be in favor of others. Thus, the factors with significant effects on the biochemical composition and nutritional quality of the sprouted grains, including biofortification and elucidation, should be examined and optimized to achieve maximum benefits to address malnutrition.

Despite their excellent nutritional benefits and potential to be included in our diet, sprouted grains are still underutilized in home and industrial food production partly due to the lack of awareness about their consumption, organoleptic properties, quality, and optimum food formulation and processing techniques. There is also a lack of technical knowledge on various industrial processing methods to enhance post-harvest shelf-life and applications of sprouted grains in food and nutraceutical products [93,94].

## 6. Recent Advances in Boosting the Nutrients of the Sprouted Grains

There is a growing interest in further enhancing the nutrients and bioactive compounds in sprouted grains by applying different interventions, as shown in Table 3. One of the simplest, easiest, cost-effective, environmentally friendly, and highly successful approaches is “agronomic biofortification”, which involves the direct addition of minerals (Zn, Se, and Fe) into the steeping water, which can be absorbed by the grains. This method has been used to increase the mineral content of staple crops as well as sprouted grains such as wheat, rice, sorghum, and pulses, especially in developing and low-income countries, to address mineral deficiency [95]. Although this method is highly successful in accumulating more minerals in the grains, it cannot improve the bioavailability of the minerals, which are essential for their functions in the body, and also less effective in increasing other nutrients. However, the combination of the biofortification and sprouting processes is highly efficient in enhancing mineral content and their bioavailability and improving the overall nutritional quality and health benefits.

Another approach to increase the nutritional quality of the sprouted grains is the use of suitable elicitors. Elicitors are materials or physical factors with biologic (biotic) or non-biologic (abiotic) origins that can induce physiological and morphological changes in the grains during sprouting to produce and accumulate more bioactive compounds and nutrients in the grains. Some of these elicitors and their effects on the nutritional quality of the sprouting grains are shown in Table 3.

Additionally, various modern technologies have been applied to promote the germination rate and uniformity and trigger biochemical and physiological reactions that enhance nutritional quality. Some of these techniques are microwave, pulse electric field, and ultrasound treatment, and their effects on the grain quality are shown in Table 3.

Biotransformation, also known as metabolic engineering, is another novel approach to enhancing the nutritional benefits of sprouted grains. It is performed by the targeted modification of cellular biochemical reactions to produce and accumulate more bioactive compounds during sprouting. Biotransformation is a new aspect of sprouted grains, which creates many opportunities for further research in the production of engineered sprouts [96].

**Table 3 foods-12-03901-t003:** Recent approaches for enhancing the nutritional quality of the sprouted grains.

Treatment	Treatment Condition	Tested Grains	Findings	References
Biofortification	Soaking in FeSO_4,_ NaSO_3_ or Na_2_SeO_3_ solutions	Wheat, Rice	Increase in Fe, Se, Zn	[4,97]
Abiotic elicitors	Water deficit stress, Inorganic salts, Metal ions, High germination temperature	Rice, Foxtail millet, wheat	Increase in protein, vitamin C, phenolic compounds, decrease in phytic acid	[98]
Biotic elicitors	Sucrose, Chitosan, Proteins, Glucosamine, Citric acid	Buckwheat, Soybean, Lentil, Wheat, Rice	Increase in flavonoids, GABA, vitamin C, B1, B2, B3 and E, proteins	[12]
Electrolysed water	Slightly acidic water with HCl, pH near neutral	Tartary buckwheat, mung bean	Suppressing microbial growth; increase in GABA	[94]
Ultrasound	50–60 Hz for 5 min	Wheat	Sanitization of fresh sprouts, increase in vitamin B2, GABA, and reduction in heavy metal contamination.	[99]
Microwave treatment	2.85 cm and frequency of 10.525 GHz for 15 min	Wheat	Increase the content of antioxidants, proteins, and amino acids, antimicrobial effects.	[99]
Magnetic field	50 mega tesla/0.5 h for 0.5, 1, and 2 h	wheat	Increase in phosphorous, potassium, and protein	[99]
Light	Light-emitting diodes light (red and blue LED)	Barely, Rice	Increase in protein and amino acid accumulation, increase in anthocyanin	[94]
Biotransformation	Cellular and DNA modifying techniques	Wheat	Increase in bioactive compounds	[96]

## 7. Final Remarks

Sprouting is a simple, sustainable, rapid, and green method to develop and accumulate bioavailable macro and micronutrients required for mitigating malnutrition and addressing various diet-related health concerns. The great features of the sprouted grains to alleviate malnutrition are summarised in Figure 4.

Diversifying the range of foods in human diets is of critical importance in tackling the global issues of malnutrition and hunger by providing essential nutrients and reducing the risk of dietary deficiencies. This approach can improve resilience to food insecurity, enhance overall nutritional intake, and promote sustainable food systems that benefit both individuals and communities. This review article offers substantial evidence supporting sprouted grains as a feasible strategy for broadening dietary options by introducing new types of nutrient-rich and readily available food resources. This approach aligns well with the UN’s action plans aimed at addressing hunger and malnutrition while reinforcing food security and building resilient supply chains.

Sprouting is beneficial for both major and minor grains. In major grains, sprouting can compensate for their poor nutritional profile. For minor grains, sprouting can be considered a value-addition strategy to boost their nutritional quality and lessen/eliminate their common issues with processing, organoleptic properties, and anti-nutritional factors. Thus, sprouting can be considered a suitable value-addition and waste-reduction strategy for underutilized grains.

It is also possible to produce tailor-made sprouts by modulating the biotic and abiotic elicitors, agronomic biofortification, biotransformation, and also applying novel technologies to design specific food applications such as malnutrition treatment, vitamin fortification, food enrichment with bioactive compounds, etc.

Sprouted grains and their derivatives have shown promising results when used in the formulation of many traditional and novel foods and beverages. However, to deliver their health benefits and mitigate malnutrition, these products should be affordable for low-income consumers who are more at risk of malnutrition and diet-related diseases.

Some main areas for future research on sprouted grains to tackle malnutrition and enhance their consumption are listed below:Identifying the most suitable grain genotype, abiotic and biotic elicitors, biofortification methods, and novel technologies for specific grains to achieve the maximum nutrients and health benefits of the sprouts;Conducting in vivo and in vitro studies to identify the unknown health benefits of fresh and processed sprouted grains;Developing environmentally friendly and economical technologies to produce shelf-stable and safe ingredients with acceptable organoleptic properties from sprouted grains for food formulation and creating novel foods.

## Figures and Tables

**Figure 1 foods-12-03901-f001:**
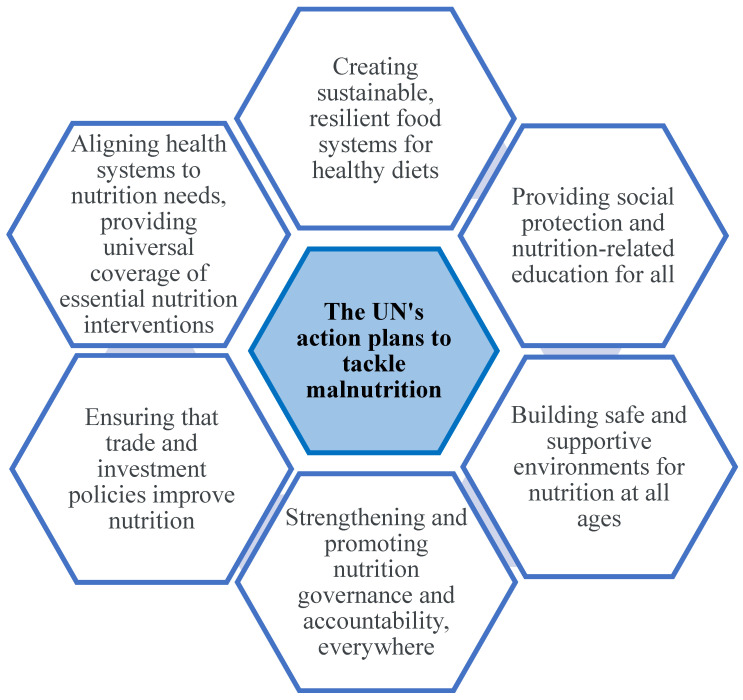
Six areas of action defined by the United Nations (UN) to tackle malnutrition (This figure is created by authors based on [7]).

**Figure 2 foods-12-03901-f002:**
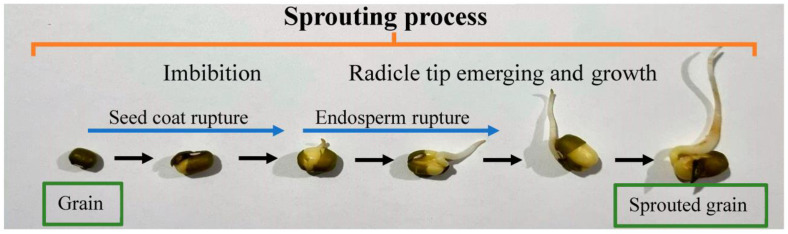
Typical sprouting process of mungbean. This figure is created by the authors.

**Figure 3 foods-12-03901-f003:**
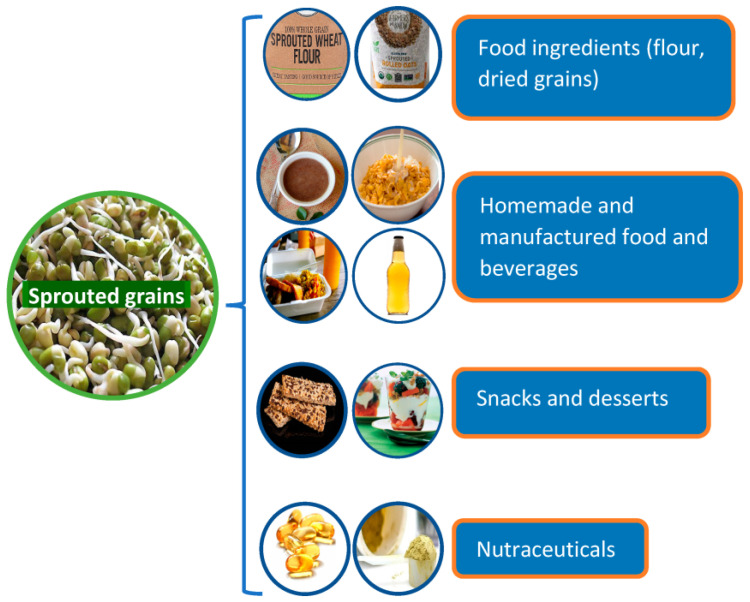
Common food applications of sprouted grains. This figure was created by the authors.

**Figure 4 foods-12-03901-f004:**
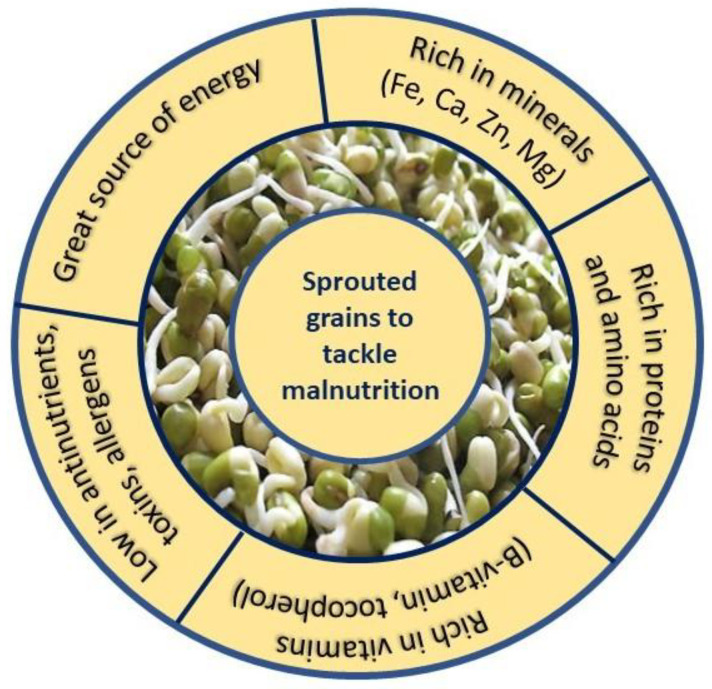
Different features of sprouted grains to combat malnutrition. This figure was created by the authors.

## Data Availability

The data used to support the findings of this study can be made available by the corresponding author upon request.

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
