# Peer review of "Unlocking the Potential of Sprouted Cereals, Pseudocereals, and Pulses in Combating Malnutrition"

_foods, 2023, doi:10.3390/foods12213901_

Round 1
Reviewer 1 Report
The manuscript is valuable scientifically, and even more so practically, because sprouted cereal grains can be an interesting, natural, nutritionally valuable addition to food.
The manuscript is quite well written. 98 well-selected literature sources were used, including nearly 50% from the last years 2020-2023. Figures and tables are of good quality and suitable for such review work.
Taking into account the following reviewer comments.
1) The title of the article „Unlocking the Potential of Sprouted Grains in Combating Malnutrition” is not consistent with the content of the article, in which the authors, in addition to sprouted cereal grains and pseudocereals, also describe the potential of sprouted legume seeds. Cereal and pseudocereal fruits are kernels, commonly called grains, while legumes (pulses) are pods that contain seeds. Therefore, the sprouted legume seeds discussed in the article cannot be classified as cereal grains and should not be included in the scope of the special issue of Foods. Moreover, the keywords include the word „grains”, which does not mean legume seeds.
2) Keywords should be written in accordance with the editorial guidelines, i.e. in lowercase letters.
3) In „*Correspondence: author: mahsa.majzoobi@rmit.edu.au” delete „author:”
4) According to the Foods, references in the text of the manuscript should be numbered in order of appearance in the text (including citations in tables and legends) and listed individually at the end of the manuscript. References should be numbered in order of appearance and indicated by a numeral or numerals in square brackets—e.g., [1] or [2,3], or [4–6].
5) In the „References” section, references must be numbered in order of appearance in the text and listed individually at the end of the manuscript. Websites cited in the text of the manuscript, e.g. https://www.who.int/news-room/fact-sheets/detail/malnutrition should be numbered and included in the „References” section.
1. Author 1, A.B.; Author 2, C.D. Title of the article. Abbreviated Journal NameYear, Volume, page range.
2. Author 1, A.; Author 2, B. Title of the chapter. In Book Title, 2nd ed.; Editor 1, A., Editor 2, B., Eds.; Publisher: Publisher Location, Country, 2007; Volume 3, pp. 154–196.
3. Author 1, A.; Author 2, B. Book Title, 3rd ed.; Publisher: Publisher Location, Country, 2008; pp. 154–196.
4. Author 1, A.B.; Author 2, C. Title of Unpublished Work. Abbreviated Journal Nameyear, phrase indicating stage of publication (submitted; accepted; in press).
5. Author 1, A.B. (University, City, State, Country); Author 2, C. (Institute, City, State, Country). Personal communication, 2012.
6. Author 1, A.B.; Author 2, C.D.; Author 3, E.F. Title of Presentation. In Proceedings of the Name of the Conference, Location of Conference, Country, Date of Conference (Day Month Year).
7. Author 1, A.B. Title of Thesis. Level of Thesis, Degree-Granting University, Location of University, Date of Completion.
8. Title of Site. Available online: URL (accessed on Day Month Year).
6) In „Author Contributions:” section, the statements recommended by the editors should be used as „Conceptualization, X.X. and Y.Y.; methodology, X.X.; software, X.X.; validation, X.X., Y.Y. and Z.Z.; formal analysis, X.X.; investigation, X.X.; resources, X.X.; data curation, X.X.; writing—original draft preparation, X.X.; writing—review and editing, X.X.; visualization, X.X.; supervision, X.X.; project administration, X.X.; funding acquisition, Y.Y. All authors have read and agreed to the published version of the manuscript.”
7) In point 49 „References”, instead of „Hubner” it should be „Hübner”
8) In the section „3. Biochemical changes” line 2: „Previous studies on the sprouted grains have indicated….” – at this point, it would be appropriate to cite a piece of literature referring to this „Previous studies….”
9) The subsection „3.2. Fibre” suggests adding „Dietary”, i.e. „3.2 Dietary Fibre”. Moreover, do the authors have data on the biochemical changes that the soluble and insoluble fractions of dietary fiber undergo during the sprouting/germination of cereal/pseudocereal grains? It would be worth including such information in this manuscript.
10) I suggests to move the entire paragraph regarding „γ-aminobutyric acid (GABA)” from subsection „3.6. Vitamins and Bioactive Compounds” to subsection „3.3. Proteins and amino acids”
11) In the subsection „3.6. Vitamins and bioactive compounds”, 2sd paragraph, line 3: the quoted sentence is not understandable „It has been found that vitamin Sprouting emerges protective responses through the synthesis and release of bioactive compounds, antioxidants, vitamins and phenolic compounds that are highly necessary for healthy body functions (Chauhan, et al. 2022, Donkor, et al. 2012, Ferreira, et al. 2019)”. What do the authors mean by „vitamin Sprouting”?
12) In the subsection „3.5. Minerals”, 2st paragraph, line 10: the sentence is not understandable „It appears that the sprouting time has a great impact on the mineral content and bioavailability of the grains” What do the authors mean by „bioavailability of the grains”?
13) In the subsection „3.7. Antinutritional factors”, maybe it would be worth adding something about oxalates as anti-nutritional compounds found in cereals/pseudo-cereals, e.g. buckwheat grains contain large amounts of oxalates, mainly calcium oxalates.
14) Figure 4 should be moved close to its citation in the text of the manuscript
The linguistic quality of the manuscript is adequate, but it is worth reviewing the manuscript for the style of some sentences.
Author Response
Thank you for your time and valuable feedback on our manuscript. Your input has been greatly appreciated and will undoubtedly strengthen our work.
Sincerely,

Reviewer 2 Report
The review article authored by Majzoobi and colleagues delves into the potential of sprouted grains from cereals, pulses, and pseudocereals in addressing global malnutrition. This article is notably intriguing from both social and scientific perspectives and is exceptionally well-written, structured, and profoundly engaging. Consequently, I highly recommend in the journal "Foods." However, there are a few minor comments that should be addressed.
Firstly, please spell out abbreviations the first time they are used in the text. For instance, the abbreviation "FDA" appears for the first time in the first paragraph of the introduction but is spelled out in the third paragraph.
Secondly, please check and correct accordingly whether the authors referred to "manganese" or "magnesium" in the fourth line after the 3.5 heading. It appears that there might be a typographical error as the symbol for manganese should be "Mn," not "Mg." A few lines below, the authors write Mg, but now it is not clear whether the authors intended to refer to "Mn" or not. Please, maintain consistency in the use of element symbols throughout the entire text.
Lastly, please ensure consistency in the list of references and adhere to the author's instructions for referencing. Some titles capitalize every word, while others do not put the Latin names of plants in italics, and there are variations in the format of DOIs, either starting with "https://doi.org/..." or simply including the DOI code. It's essential to follow a uniform style for references throughout the article.
Author Response

(The authors gave the same response as above.)
